physiology

metabolism, winter adaptation,
oxygen consumption, Dehnel's Phenomenon

**Author for correspondence:**
Paul J. Schaeffer
e-mail: schaefpj@miamioh.edu

†Shared first authorship.

# Metabolic rate in common shrews is unaffected by temperature, leading to lower energetic costs through seasonal size reduction

Paul J. Schaeffer[1,†], M. Teague O'Mara[2,3,4,†], Japhet Breiholz[3,4], Lara Keicher[3,4], Javier Lázaro[3,4], Marion Muturi[3,4] and Dina K. N. Dechmann[3,4]

[1]Department of Biology, Miami University, Oxford, OH 45056, USA
[2]Department of Biological Sciences, Southeastern Louisiana University, Hammond LA 70402 USA
[3]Department of Migration, Max Planck Institute of Animal Behavior, am Obstberg 1, 78315 Radolfzell, Germany
[4]Department of Biology, University of Konstanz, 78462 Konstanz, Germany

PJS, 0000-0001-6230-8121; MTOM, 0000-0002-6951-1648; LK, 0000-0001-5051-5588; JL, 0000-0003-1958-6470; DKND, 0000-0003-0043-8267

Small endothermic mammals have high metabolisms, particularly at cold temperatures. In the light of this, some species have evolved a seemingly illogical strategy: they reduce the size of the brain and several organs to become even smaller in winter. To test how this morphological strategy affects energy consumption across seasonally shifting ambient temperatures, we measured oxygen consumption and behaviour in the three seasonal phenotypes of the common shrew (*Sorex araneus*), which differ in size by about 20%. Body mass was the main driver of oxygen consumption, not the reduction of metabolically expensive brain mass. Against our expectations, we found no change in relative oxygen consumption with low ambient temperature. Thus, smaller body size in winter resulted in significant absolute energy savings. This finding could only partly be explained by an increase of lower cost behaviours in the activity budgets. Our findings highlight that these shrews manage to avoid one of the most fundamental and intuitive rules of ecology, allowing them to subsist with lower resource availability and successfully survive the harsh conditions of winter.

# 1. Introduction

For animals living at higher latitudes, seasonality poses a severe challenge to survival. Cold temperatures increase metabolic costs even during rest [1,2], and this makes winter challenging especially for small endothermic vertebrates. Their mass-specific metabolism is much higher than that of large animals and they have a particularly high need for strategies to cope with winter conditions [3,4].

In response to seasonal environmental change, countless species reversibly modify their adult phenotype, including behaviour, morphology and physiology [5]. This phenotypic flexibility can act on several energy saving pathways to ensure winter survival. The metabolic theory of ecology (MTE, [1]) provides an important framework to tease apart the effect of several parameters, which may play a role here. It states that temperature and body mass are the main drivers of energetic costs. In endotherms, decreasing temperature leads to increased metabolic heat production to maintain body temperature, although this effect can vary considerably in animals adapted to cold versus warm climates or of different body size [1,2,6]. Body mass also plays a large role in determining metabolic rate [7–9], but variation in metabolic rates may be significant and related to several ecological or evolutionary factors (reviewed in [10]). Similarly, reduced conductance can reduce metabolic demand as the need for heat production is reduced. Conductance also scales predictably with body size, but increases at a lower rate than metabolism [11]. Given the small size of shrews, high relative conductance is a potential challenge to winter survival. Reducing conductance is particularly challenging for very small animals given that increases in fur thickness necessary are very difficult in such a small animal ([12] but see [13]). With already high metabolic rates and high conductance only partly buffered by winter fur, reducing body size may be an alternative adaptive response to dealing with energetic challenges imposed by winter in small mammals that cannot migrate away or hibernate to avoid the energy costs of maintaining body temperatures at cold ambient temperatures ([14], but see [15]).

Many species lose mass (mainly fat) over the course of the winter [16], but there is mixed support for the argument that this decreases overall energy requirements. In male weasels, body mass can vary from 45 to 150 g within a single population, and this variation has a strong effect on over-wintering strategies [17]. In agreement with the MTE, body mass and temperature are the factors that best explain variation in inter-individual energy consumption in weasels [18]. However, ideal body mass can vary, even over the course of winter. Similarly, in root voles, individuals with the highest metabolic capacity survive best, but in addition, small-bodied individuals have a higher probability of survival early in the winter, while larger bodied individuals survive better at the end of winter [19].

An example of a rare, but intriguing alternative wintering strategy is Dehnel's Phenomenon. This seasonal reversible size change has only been described in small, non-hibernating red-toothed shrews and some mustelids [20–22]. In the common shrew (*Sorex araneus*), Dehnel's Phenomenon is reflected by an up to 25% size change of the skull, 10–20% change in the mass of the brain, large changes in the mass of several major organs, as well as overall body size [23–26]. Importantly, in *S. araneus*, this reversible size change has been followed within individuals across seasons [23], validating this system for studies of both the effects of size changes and external influences on metabolic costs of living. *S. araneus* are born in summer, rapidly grow up to a maximum first size (summer juveniles), then begin to shrink in early autumn and reach their minimum in February (winter subadults). Soon after, they begin to grow again and sexually mature (spring adults), after which they reproduce and die shortly thereafter. Final body mass of spring adults exceeds juvenile pre-shrinkage mass, while some tissues, including the brain, only partially regrow [21,25,27]. Researchers have hypothesized that winter shrinkage results in a reduction of total energy consumption due to the reduction of energetically expensive tissues, such as the brain [4,28,29]. In support of this, overall energy consumption at rest in thermoneutrality is 18% lower in size-reduced winter animals compared to the larger summer juveniles, likely due to a combined effect of winter fur insulation and smaller size [13]. However, one of the biggest challenges of winter for a small non-hibernating animal should be the greatly reduced temperatures leading to high potential cost of thermoregulation, which should be enhanced by the less favourable surface to volume ratio in the smaller winter phenotype.

Several species of *Sorex* have low summit to basal metabolic rate ratios [30], suggesting that they always operate close to their maximal energetic capacity due to this low metabolic scope. They seem unable to rely on seasonal modulation of mass-specific energy expenditures in response to environmental variation. In addition, *S. araneus* have higher energy expenditure than predicted by size alone [31]. A normothermic 10 g *S. araneus* has the same basal metabolic rate as a sympatric 25 g vole [32] and nearly twice that of closely related and similarly sized, sympatric white-toothed shrews [4]. Whether this energy expenditure increases or decreases with the size change at harsh ambient winter conditions is

thus extremely relevant for the understanding of this phenomenon. Alternatively, winter may not be the most energy-demanding season for *S. araneus*. Both sexes face extremely high additional energetic demands during the reproductive season in spring and summer when they become sexually mature. They produce nearly double the number of pups as similarly sized white-toothed shrews, while their gestation period is shorter and the weaning period is longer [33]. Lactating *S. araneus* increase their food intake much more than white-toothed shrews, with feeding rates up to 2.5 times higher than their own winter intake [4]. Both sexes of *S. araneus* are territorial and solitary and they increase territory size during the spring growth period. During this time body mass, but not brain mass of both sexes is highest.

Even if it were energetically less costly, smaller size may also be risky. *S. araneus* have little ability to store fat [34], and in winter, they turn over their body fat available to metabolism within about 4-5 h [35] leading to multiple activity cycles per day [31,36]. They are active during 58% of the day [37] compared to 25–30% in white-toothed shrews [38]. Several non-hibernating animals are known to reduce activity during winter (e.g. weasels [18], and Hottentot golden moles [15]), and at least in weasels, this appears to lower seasonal energy costs. It is unknown if the already high level of activity of *S. araneus* varies by season, and if this has significant effects on daily energy expenditure.

To tease apart the influence of mass, temperature and activity, we first compared the rate of oxygen consumption between animals at each stage of Dehnel's Phenomenon to assess the effects of changing body size. Very importantly we made these measurements outside at ambient environmental temperatures (see also [39]) to assess the relative importance of size changes under natural temperature variation. We hypothesized that if temperature has a strong effect on metabolic rate, then both relative and total oxygen consumption should be higher in winter subadults than in the other two groups regardless of size due to thermoregulatory costs. We generated daily activity budgets during each stage with continuous video recording and hypothesized that the relative amount of time and energy spent with expensive behaviours would decrease in winter, reducing overall energy costs. With small size and intense activity, *S. araneus* are an excellent model to determine the impact of individual size changes on seasonal energy consumption as well as the importance of behavioural plasticity in modifying energy demand.

# 2. Material and methods

## 2.1. Trapping of animals

We trapped 29 common shrews (*S. araneus*, Linnaeus 1758) between 0700 and 1100 in the forest and meadows around Möggingen, southern Germany (longitude 8.994, latitude 47.766) in June (large juveniles, $n = 12$), January/February (size-decreased subadults, $n = 9$) and April/May (regrown spring adults, $n = 8 + 1$ recapture from January) of 2016 and 2017 with wooden live-traps (Jerzy Chilecki, Białowieża, Poland) containing nesting material, baited with mealworms and checked at 2 h intervals. New animals were captured during each season, as recapture rate between seasons is only about 10% [23], which would have required measuring several hundred juveniles to ensure that 12 were measured throughout the cycle.

## 2.2. Selection and measurement of study animals

We classified individuals as juvenile, subadult or adult based on the annual life cycle of the shrews and external morphological characteristics [21,40,41]. Palpably pregnant females were excluded. We weighed each shrew immediately upon capture and then held shrews individually in a cage within an outdoor aviary with two connected chambers, one with deep soil, bedding and nesting material, the other with a shallow soil layer, a feeding dish, water *ad libitum* and a running wheel [42], until the respirometry experiments began. Individuals were weighed to ±0.01 g (Sartorius U5000D; Göttingen, Germany) just before they were transferred to the respirometry chamber (see below).

Shrews captured in April and June were first used in a study of carbon isotope turnover at which breath samples at specific times spanning 96 h were taken [35]. During this period, shrews were fed their body weight daily with a chicken/cricket mixture and removed temporarily from their cages for approximately 5 min to conduct short breath sampling, results not presented herein. Thus, respirometry experiments were done about 4 days after capture. For shrews captured and measured in January/February, only a single breath sample was taken just before they were transferred to the respirometry chamber (within one day of capture in all cases).

After measuring oxygen consumption, we X-rayed the shrews under anaesthesia in an induction chamber (Surgivet, Dublin, USA; oxygen flow rate $1\,l\,min^{-1}$, 5% isoflurane) connected to a Titus anaesthesia System (Dräger, Lübeck, Germany). We placed anesthetized individuals into a form-fitting foam bed to ensure standardized body position. We X-rayed animals in a Faxitron MX 20 cabinet (26 kV, 6 s using an OPG Imaging Plate, Gendex, Hatfield, USA) and extracted the images with a scanner (DenOptix, Gendex, Hatfield, USA). We took both ventral and lateral X-ray images of the skull. We measured skull height on the electronic X-ray files in Image-J as the distance from the tympanic rings to the dorsal surface of the braincase. All X-ray measurements were taken blind regarding capture date by a single observer to avoid bias. We size corrected skull height by the maxillary tooth row length as this parameter does not change seasonally within an individual [23]. We released shrews at the place of capture after a maximum of 2 h of recovery.

## 2.3. Oxygen consumption

We measured the oxygen consumption rate of each shrew in continuous open-flow respirometry over a 12 h period. The start time for each run was randomly distributed to attempt to span an entire, representative day and not make the respirometry sessions too long for individuals. Respirometry was done under environmental ambient temperature and light conditions with the shrew in a sealed plastic box $(300 \times 180 \times 170\,mm)$ placed outdoors, containing nesting material (fleece and hamster wool), a running wheel and two small dishes for food and water. We used white sand as substrate to enhance contrast for video recordings. To eliminate respiration by microorganisms, the sand was dried and autoclaved, and fresh sand was used for each animal to avoid transference of any scent markings. In addition to the incurrent and excurrent air ports, a third hole was drilled into the lid and a syringe barrel with the end removed was mounted into the hole. This was sealed by the plunger and brief removal of the syringe plunger permitted us to feed the shrew, which was done every 1.5 h (0.8 grams per event) throughout the respirometry trial to mimic natural activity cycles and food availability to a wild shrew. Air was pushed first through a Drierite column to pre-dry the air, then through a mass flow meter (SIERRA Mass Flow Controller, Sable Systems, USA), which set total flow to $1.0\,l\,min^{-1}$. After the chamber, air again passed through a Drierite column, followed by an FC-1B oxygen analyser (Sable Systems, Las Vegas, USA). Analogue outputs of the flow meter and the $O_2$ analyser were converted into a digital output with a UI-2 (Sable Systems) and recorded using ExpeData software (Sable Systems). We used an oxygen pulse to determine the response time of the system and found the delay between injection of the pulse and detection at the analyser to be 35 s. We used this delay time to synchronize the respirometry data with the video data, which we obtained simultaneously (see below).

We calculated the rate of oxygen consumption for each second (except during feeding each 1.5 h as this introduced noise in the recording), using the following equation from the ExpeData manual,

$$VO_2 = V_E \cdot (F_iO_2 - F_eO_2)/(1 - F_eO_2 \cdot (1 - RQ)) \tag{2.1}$$

where $V_E$ is the total flow in the chamber (in $ml \cdot min^{-1}$), $F_iO_2$ and $F_eO_2$ are the fractional content of oxygen in incurrent and excurrent air, respectively, and $RQ$ is the respiratory quotient which is assumed to be 0.85 in these analyses, a value consistent with a mixed carbohydrate, fat and protein diet.

## 2.4. Video and behaviour analysis

To record the behaviour of the shrew throughout each 12 h respirometry trial, a DV-883.IR video camera (Somikon, Germany) was mounted above the cage. For those animals, whose 12 h measurements were made during the night, two red lamps were also placed above the cage. We coded behaviour directly from video in 'CowLog' (version 3.0.2; Natural Resources Institute, Helsinki, Finland). We distinguished five classes of behaviour: 'Rest', 'Eat', 'Drink', 'Run', and 'Walk'. Run was reserved for use of the running wheel, whereas Walk denotes other activity, such as exploring the cage, climbing on the running wheel, or building a nest. Eat was only coded when it was clearly observed, and Eat and Drink were grouped together for analysis. Behavioural codes were applied to each second of the experiment, except when food was delivered to the chamber. Because of technical issues, there were gaps in the video data for two animals in April which reduced the sample size in April to $n = 6$. The sample size in June was $n = 9$ and in January $n = 8$.

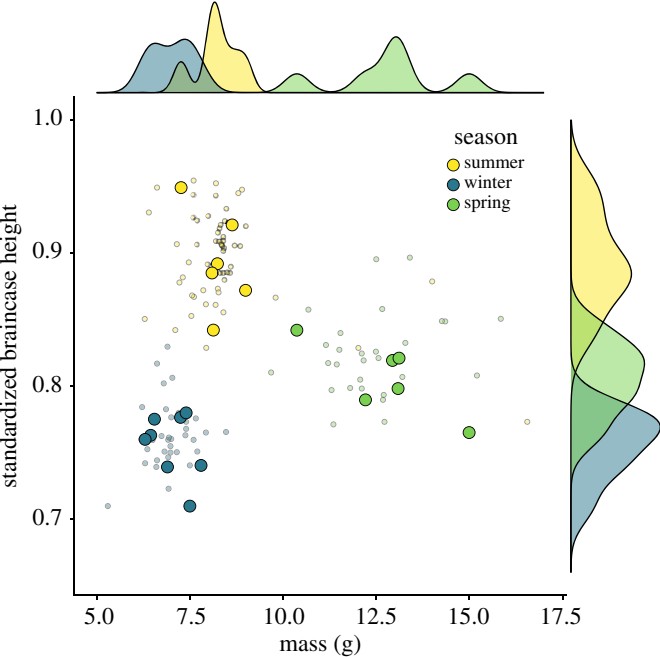

**Figure 1.** Shrew mass (g) and standardized braincase heights measured from X-rays. All capture data from 2014 to 2018 are shown in small circles, and the individuals presented in this study are shown as large circles. Both shrew body mass and braincase height differ by season for the animals in this study.

## 2.5. Statistical analysis

The 1 Hz respirometry data were aggregated over 180 s (*ca*. 5 × system time lag) and an overall mean was calculated. This resulted in 482–696 observations per individual. We were interested in testing for the effects of body mass, standardized skull height, season and temperature on both whole animal (total) and relative (mass-specific) metabolic rates. These aggregated data were entered into generalized linear mixed effects models in *nlme* [43] fit via REML that used individual identity as a random intercept and season as a random slope, and a continuous AR1 (*corCAR1*) correlation structure to account for the inherent temporal autocorrelation in any metabolic measure [44]. Since the continuous predictors of interest (body mass, skull height, temperature) occur over different ranges, we scaled each variable over its full range using the *scale* function. Dehnel's Phenomenon is a complex process, and we used model selection to understand how our variables of interest affect oxygen consumption. We built models in a stepwise progression of single variables and of two-way interactions. We evaluated the significance of all factors via *nlme::anova.lme* and also calculated the Akaike Information Criterion corrected for small sample sizes in *MuMIn::AICc* to identify the best fitting models [45]. All data shown have been back-transformed accordingly, and all analysis was conducted in R 3.5.1 [46].

## 3. Results

To assess the seasonal size changes of our three cohorts, we compared body mass and braincase height in summer juveniles, winter subadults and spring adults. Similar to our previously published results [23,27], we found that season had a significant effect on both size variables (mass: $F_{2,19} = 77.02$, $p < 0.001$; braincase height: $F_{2,18} = 77.02$, $p < 0.001$, figure 1). Summer juveniles ($n = 6$) were smaller than spring adults ($n = 6$), while winter subadults ($n = 8$) were the smallest, showing the seasonal size reduction characteristic of the Dehnel's Phenomenon [20,21]. Braincase height (standardized to tooth row) was also smallest in the winter subadults. Regrowth in spring adults resulted in higher skulls, but these were proportionately smaller than those of summer juveniles. The data from the animals in this study are also overlaid with a larger, previously published dataset [23,27] indicating that these shrews were representative of the larger population (figure 1).

Shrews could alter their activity to minimize energetic expenditure across the seasons. Activity budgets were calculated per individual as percentage of total observation time engaged in Eat + Drink, Rest, Run or

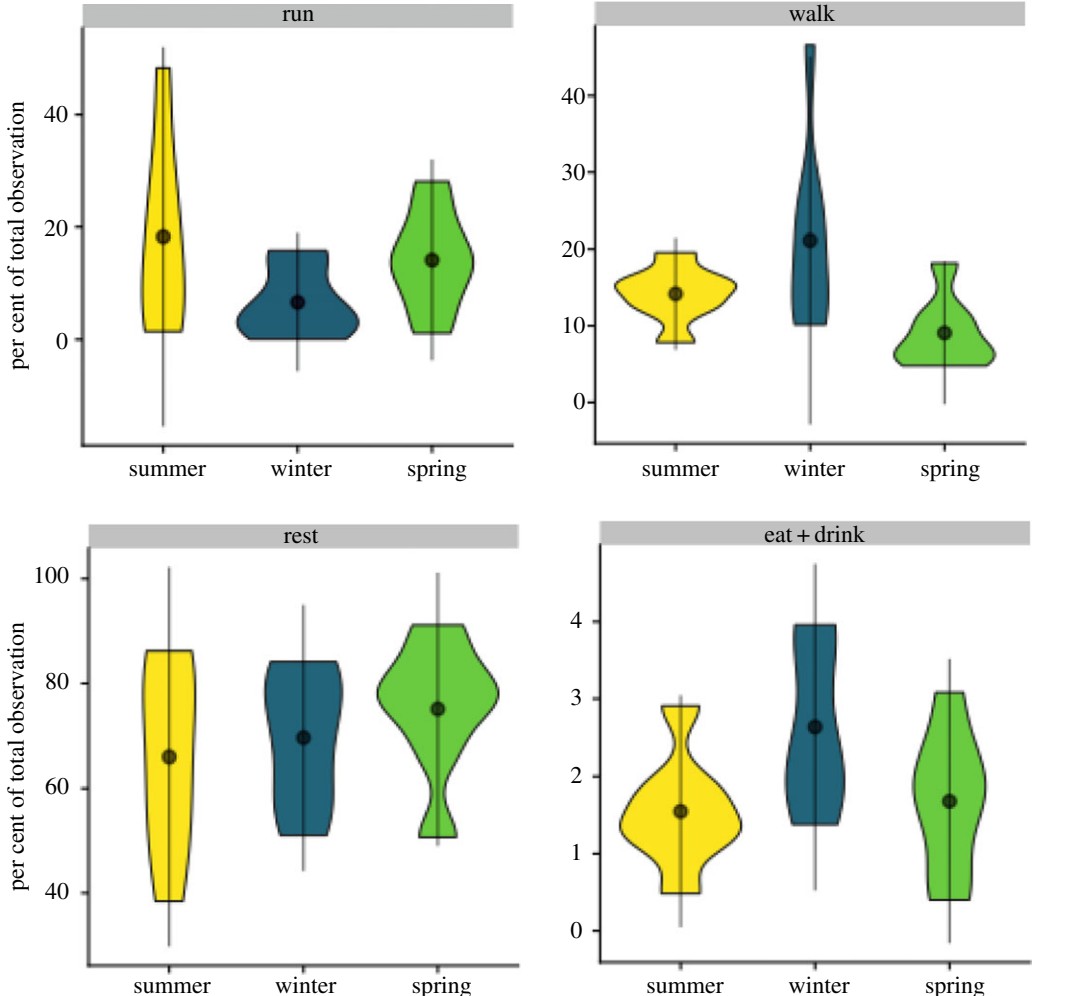

**Figure 2.** Activity budgets as per cent of total observations that shrews spent in four activities. Shrews in winter spend more time eating and looking for food.

Walk during the respirometry trials and these differed among the seasons (figure 2, $F_{11,76} = 57.77, p < 0.001$). Summer juveniles and spring adults showed the same activity distribution. In winter, shrews spent less time running than in either summer or spring, and more time eating, drinking and walking.

We report both whole animal (total) and mass-specific oxygen consumption measured with flow-through respirometry over 12 h at ambient temperatures, with the animals moving freely in a cage provided with a nest, a running wheel, ad libitum water and food at regular intervals. Whole animal oxygen consumption rate reflects the total energy expended at each of the three life stages of summer juvenile, winter subadult and spring adult, and thus the energy intake required to meet that demand, while mass-specific oxygen consumption rate can reveal adaptation to increased demand. The best models for whole animal oxygen consumption rate revealed a strong effect of body mass, while ambient environmental temperature and season were also significant as individual effects (table 1). However, when body mass was entered into these models, the effects of all other variables became insignificant. The AICc model selection (electronic supplementary material, table S1) shows that the mass-only model best fits the metabolic rate data. As expected, increasing body mass increased whole animal oxygen consumption (figure 3a). Surprisingly, decreasing temperature was associated with slightly lower oxygen consumption across seasons. However, within each season, there was a slight increase in oxygen consumption with decreasing temperature (figure 3c). The apparent reduction in the intercept in summer juveniles and, especially, winter subadults suggests a resetting of the relationship between temperature and whole animal oxygen consumption at each developmental stage. By contrast, we found no single or interactive predictive factors for mass-specific oxygen consumption. The best-ranked model included the interaction between temperature and scaled braincase height, but no factor approached a significant relationship across season, and mass-specific oxygen consumption remained fairly consistent across seasons (figure 3d–f).

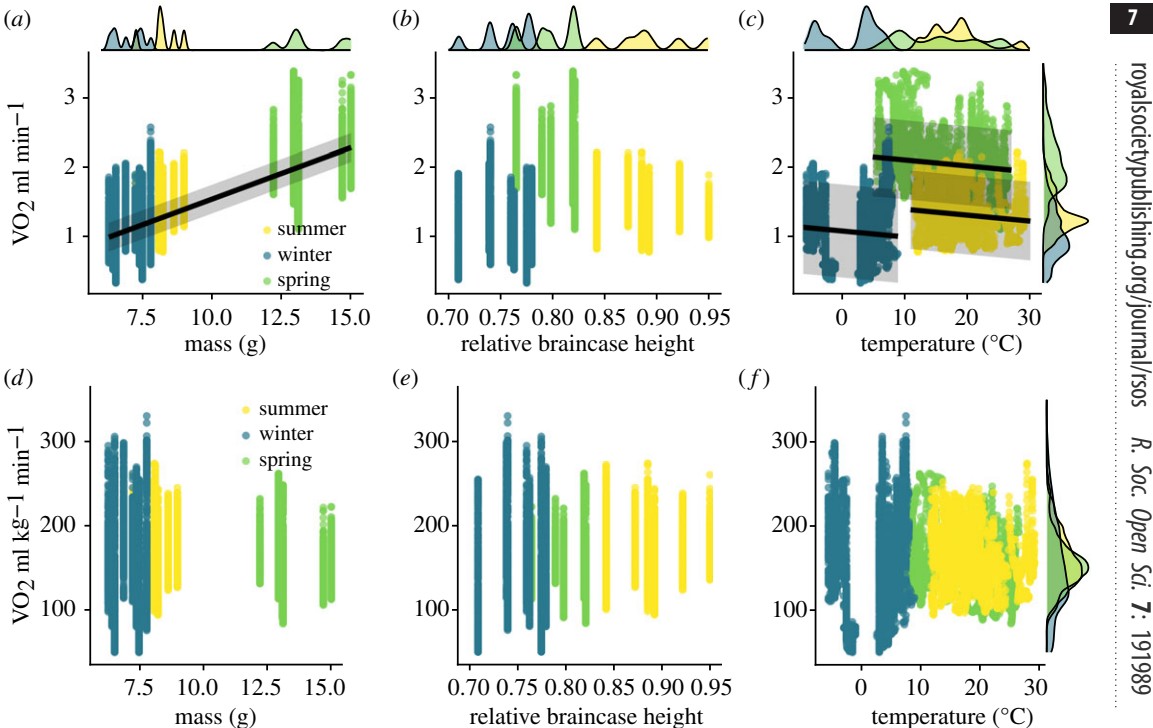

**Figure 3.** Shrew absolute metabolic rates (a–c) and relative metabolic rates (d–f) predicted by body mass, relative braincase height and ambient temperature. Margin plots show the density of each variable in the row or column. Mass, season and temperature had significant effects on absolute metabolic rates on their own and the lines in (a) and (c) show the predicted relationship from the models that account for random slope of season, random intercept of individual and a continuous time autocorrelation structure within the data.

**Table 1.** Best fit models predicting total metabolic rate and relative metabolic rate for shrews across three seasons. Models included individual identity as a random intercept, season as a random slope and a temporal autocorrelation structure. The full model selection is given in electronic supplementary material, table S1.

| | AICc | factors | estimate ± s.e. | T [DF] | p | anova F [DF] | anova p-value |
|---|---|---|---|---|---|---|---|
| total MR ~ mass | −34226.92 | (intercept) | 1.42 ± 0.03 | 45.17 [13262] | | | |
| | | mass | 0.42 ± 0.03 | 13.38 [18] | <0.001 | 179.07 [1, 18] | <0.001 |
| relative MR ~skull height * temperature | 82949.02 | (intercept) | 157.54 ± 5.65 | 27.88 [12221] | | | |
| | | skull height | 9.98 ± 5.56 | 1.8 [17] | 0.09 | 1.83 [1, 17] | 0.19 |
| | | temperature | −5.9 ± 5.05 | −1.17 [12221] | 0.24 | 1.06 [1, 2221] | 0.3 |
| | | skull height* temperature | −3.23 ± 5.52 | −0.59 [12221] | 0.56 | 0.34 [1, 2221] | 0.56 |

## 4. Discussion

Winter energy demands of endotherms are generally assumed to be among their highest across the annual cycle given the challenges of thermoregulation. These high costs can exceed those of reproduction in both

mammals [42] and birds [47,48], and, with low food availability, presumably drove the evolution of hibernation and migration in many species. For small endotherms, such as shrews, their large surface area relative to body mass should make them particularly vulnerable to heat loss [2]. In spite of this, shrinking the whole body as well as several major organs has long been suspected to result in energy savings although the underlying mechanisms were unclear [4,25,28,29]. We expected both body mass and environmental temperature to influence organismal metabolic rate [1]. Instead, the effect of temperature on oxygen consumption in shrews housed at ambient environmental temperature was opposite to our predictions. Size reduction in shrews reduced absolute energy consumption and thus overall energy costs across a wide range of ambient temperatures.

Previous data had shown constant mass-corrected energy consumption in summer juveniles and winter subadults in thermoneutrality [13]. Although we found a slight elevation of energy expenditure with decreasing temperature within each season, absolute oxygen consumption decreased with decreasing temperature across seasons. Shrew oxygen consumption seemed to ignore temperature, leaving only mass effects to affect winter energy demand. This may be partly due to decreased conductance, which, similar to several, much larger arctic mammals and birds [9] differ only by about 30% between summer juveniles and winter subadults in *S. araneus* [13]. Fristoe *et al.* [11], who found that conductance in shrews is close to that predicted by allometric relationships and thus higher, did not compare this between sizes within a species.

However, this only partly explains why winter temperatures did not affect energy consumption in *S. araneus*. Our additional hypothesis was that these shrews should reduce high cost behaviours in winter (see also [18]). However, red-toothed shrews have a constant need to seek out prey in the wild (*ca.* every 90 min [49]), and they are not known to use torpor [4,33]. We believe this is reflected in the activity budgets we observed. Our winter subadults allocated their time differently than shrews in the other seasons. They spent less time running and more time walking and feeding. It is likely that they were investing more time in foraging to meet higher energetic and nutritional needs, due to increased fat turnover rates in winter [35]. All of our shrews spent more time at rest then what was observed in previous work (summer: $66 \pm 18\%$, winter: $69 \pm 13\%$, spring: $75.1 \pm 13\%$, figure 1) [49]. Winter animals spent more time at rest than the summer juveniles, but less than spring adults (figure 2), which nonetheless had the highest energy expenditures due to large body size (figure 3*a*).

A common assumption is that winter energy expenditure in animals that show Dehnel's Phenomenon [4,28,29] is affected by the size reduction of expensive tissues, especially the brain. Our results were not consistent with brain size as a main determinant of oxygen consumption. Our experimental animals followed the previously described pattern regarding the relationship between body mass and brain size (as represented by skull size, figure 1; see also [23]). Brain size was largest in the summer juveniles, but energy consumption was highest in the heaviest phenotype with intermediate brain size—the spring adults. The importance of body mass is reinforced in the relative metabolic rate models as no combination of our tested variables can explain a significant amount of variation. However, it is interesting that the best model to describe relative oxygen consumption includes the interaction between relative skull height and temperature. Thus, while the reversible change in the size of the energetically expensive brain and in cognitive function [50] remains intriguing, it is not a primary driver of Dehnel's Phenomenon (figure 3*e*). The brain is likely a target of reduction as overall metabolic rates are decreased. It is important to note that the rest of the size reduction is not simply a reduction in adipose mass; the brain is only one of several organs and tissues changing size. Further work is needed to sum up the cumulative effect of all these changes to assess their role in the overall energy consumption of the shrew as it goes through Dehnel's Phenomenon, as the size of peripheral organs has been found to influence overall energy consumption in birds [51]. Somewhat paradoxically, reduction in size further reduces the already extreme surface area to volume ratio in common shrews, which would be predicted to increase the rate of heat loss in winter and thus the demand for metabolic heat production.

The mechanisms by which our shrews were able to defy the predictions of the MTE regarding the effects of temperature remain unclear. Since Scholander *et al.* [2], it is generally assumed that endotherms increase heat production and thus metabolic rate below a critical temperature. We would have expected a several-fold increase in metabolic rate solely from the temperatures our shrews experienced, which reached well below zero. However, thermal substitution of the heat production with activity and/or the heat increment of feeding can reduce or even eliminate the costs of thermogenesis [52-55]. We suspect that due to their high metabolism common shrews constantly produce excess amounts of heat. This is thought to be particularly efficient in medium-sized mammals and birds at intermediate temperatures, but recently hummingbirds have been found to produce so much muscle heat that hovering is neutral in terms of thermoregulatory costs [56,57]. This indirectly serves to keep their body temperature high and

they likely face a larger themoregulatory problem in summer than winter, supported by the great success of this genus in terms of numbers of species as well as individuals at high latitudes [4]. Smaller body size in winter then simply reduces their food requirements allowing them to subsist on lower amounts or on lower quality of food in winter [5,33], resulting in an animal which against all odds seems to do as well if not better in winter.

Ethics. All shrew handling and sampling methods were approved by the Regierungspräsidium Freiburg, Baden-Württemberg (permit no. 35-9185.81/G-15/128).

Data accessibility. Data available from the Dryad Digital Repository: doi:10.5061/dryad.98th04m.

Authors' contributions. D.K.N.D., J.B. and P.J.S. conceived and designed the study; J.B., L.K., M.M., J.L. and P.J.S. collected the data; M.T.O., P.J.S., J.L. and J.B. analysed data; D.K.N.D., M.T.O. and P.J.S. wrote the manuscript with contributions from all authors. All authors gave final approval for publication.

Competing interests. We have no competing interests.

Funding. This work was funded by the Max Planck Poland Biodiversity Initiative to D.K.N.D.

Acknowledgements. We wish to thank Ina Köchling and the animal caretakers of the Max Planck Institute of Animal Behavior for help during the fieldwork and experiments. We also want to thank our two anonymous reviewers for their helpful comments.

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
