## [Reviewer comments · Royal Society Open Science]

Review History

RSOS-191989.R0 (Original submission)

Review form: Reviewer 1

Is the manuscript scientifically sound in its present form?

No

Are the interpretations and conclusions justified by the results?

Yes

Is the language acceptable?

Yes

Do you have any ethical concerns with this paper?

No

Have you any concerns about statistical analyses in this paper?

No

Recommendation?

Accept with minor revision (please list in comments)

Comments to the Author(s)

This paper investigates changes in metabolic rate in response to seasonal changes in temperature and phenotypic changes in body size and relative brain size to further elucidate the mechanisms behind Dehnel's Phenomenon. The methods and data are sufficiently executed. The interpretation of the results are adequate.

Abstract: Generally good. I would suggest including a brief explanation for how the authors ruled out the following in the abstract "Body mass was the main driver of oxygen consumption, not the reduction of metabolically expensive brains". For example, through the use of generalized mixed effects models. This is also unclear in the objectives paragraph 110-120.

Introduction: The authors state that MTE provides an important framework to tease apart the effect of body size and temperature as the main energetic costs for endotherms. This isn't exactly true. The temperature term in MTE (Brown et al. 2004) is included to accommodate ectotherms that vary in body temperature in response to ambient temperatures. The major extension of MTE and Scholander-Irving to accommodate impacts of temperature in endotherms to body size and metabolic rate comes from Fristoe et al. (2015 PNAS). That and Scholander showed how changes in environmental temperature influences changes in body temperature via metabolic rate and conductance. It seems the alternative hypothesis the authors are testing is that the mass-specific metabolic rate would be greater for body size in winter to offset heat lost to cold, although the authors do not find support for this. This may inform predictions in changes in body temperature, which the authors do not report here (and that's OK). But may wish to include in future studies. But I do think this section of the introduction should be revised to clearly explain the link between MTE and predictions.

Line 45: "It states that temperature" do the authors mean environmental or body temperature or both? See above explicit distinction in theory. This should be clear throughout when temperature is used, for example, in lines 58-59 is it body or environmental temp?

Line 85: Are the author's referring to metabolic scope?

Line 89: If *S. araneus* at 10 g has the same BMR as 25 g vole and two that of a similar sized white-toothed shrew, the does it have a higher body temperature, higher conductance or operate and colder environmental temperatures (e.g., Fristoe et al. 2015)? The authors do not report body temperature or conductance, but these links need to be clear in the context of the theory.

Line 113: again, you must be referring to environmental temperature unless you measured body temperature too?

Line 232: Please reference previously published studies you refer to.

Line 239: Please cite the previously published data source.

Paragraph 312-333: is very long. Consider breaking in to two.

Review form: Reviewer 2**Is the manuscript scientifically sound in its present form?**

Yes

Are the interpretations and conclusions justified by the results?

Yes

Is the language acceptable?

Yes

Do you have any ethical concerns with this paper?

No

Have you any concerns about statistical analyses in this paper?

No

Recommendation?

Accept with minor revision (please list in comments)

Comments to the Author(s)

General: I found the study as an interesting and thought-provoking. Authors continuously produce papers discussing morphological and physiological adaptations to seasonal changes of ambient temperatures in small, non-hibernating mammals (e.g. Dechmann et al. 2017, LaPoint et al. 2017, Lázaro et al. 2017 and 2018). Here, along with changes in body mass and skull size in common shrews across three seasons (summer, winter and spring), Authors estimated individual's energy consumption (absolute and mass-specific) together with five classes of behavior activity (rest, eat, drink, run and walk). And these two last measurements, metabolic rate and behavior, add new facts to the number of papers testing animals adaptations to challenging environmental factors. The demonstration of seasonal phenotypic plasticity in morphological, behavioral and metabolic traits is fundamental not only for better understanding one of winter strategy in small mammals called Dehnel's phenomenon, but also for the whole evolutionary physiology. From this perspective the study is important. However, I have several comments, which can help Authors make some corrections and improve the paper.

ABSTRACT

Page 2, line 30: This conclusion should be more specified. Activity is an important component of energy budget, and it is reflected in oxygen consumption. Please, re-phrase this sentence to make it less general.

INTRODUCTION

I have a little confusion with the expression of the aim of the study. If my meaning of Authors' intention is correct, scientific reasons for which the work was done are presented on page 5, lines 91-92 and lines 108-109. But now, they are hidden in the main text. Both arguments should be strongly emphasized together with a general description of methods used for such comparisons. It ones read then oxygen consumption was measured outside at ambient environmental temperatures (lines 111-112), but no word about a way in which a class of behavior was recording. Please, re-write lines 110-114 to make that paragraph (very essential for Introduction) more informative and scientific important.

Pages 5-6, lines 114-120: Re-think and make hypotheses more clear. In Discussion there is the statement 'we expected both body mass and environmental temperature to influence organismal metabolic rate' (page 12, lines 286-287). I can't see any detailed speculation about an effect of body mass reduction in shrews on their metabolic rate in that part of Introduction.

Page 4, line 84: Shouldn't be 'surface to volume ratio'?

METHODS

Page 6, line 140: Put information how long shrews have been waiting for the beginning of the experiments. The time course between capturing and experiments was the same for individuals from each season or not? If not, way?

Page 6, lines 141-144: I have mixed feelings with that information. It describes the method for incorporation digested food into metabolism and assessment of turnover rate of body fat, which was one of the aim in previous Authors' study (I suppose it is Keicher L., O'Mara M.T., Voigt C.C., Dechmann D.K. 2017. Stable carbon isotopes in breath reveal fast incorporation rates and seasonally variable but rapid fat turnover in the common shrew (*Sorex araneus*). *J. Exp. Biol.* 220, 2834-2841). Moreover, it looks like the same group of shrews was used in the previous paper and this work. And of course, a way like this is acceptable in results sharing and publishing, but I can see no scientific need for putting information like that here. Results on incorporation rates and fat turnover were published and they are not presented in actually reviewed paper. I suggest removing that information.

Page 6, line 144: Phrase 'two and seven days after capture' is confusing. It is not clear to which step of experiment it relates to. And again, does it describe the period of time before the start of oxygen consumption analyses? Please, specify it.

Page 7, line 147: Remove the phrase 'for 12 hours'. It is mentioned below, in line 164 (and refers to all shrews used in measurements of oxygen consumption).

Page 7, line 157: Please decide which form 'braincase' or 'brain case' is correct and apply it throughout the paper.

Page 7, lines 159-160: How long recovery time have been lasting before releasing shrews at their place of capture?

Page 8, lines 198-200: For each individual the 12-hour respirometry trial was conducted within a day, while behavior recording took 24 hours (including a night)? Am I right?

RESULTS

Page 10, line 240: Cite an appropriate paper at the end of the sentence.

Page 10, lines 247 248: Shouldn't be 'spring adults' instead of 'winter subadults'?

DISCUSSION

Page 12, lines 308-309: Cite an appropriate paper.

Page 13, line 319: Cite adequate figures.

Page 13, line 324: Put 'Figure 3E' instead of 'Figure 3D'.

OTHER COMMENTS:

Figure 1 and 2. The sample size for winter subadults and spring adults on both figures is the same (n=8 and n=6, respectively). Explain the difference in the number of summer juveniles presented on Figure 1 and 2 of (in Methods, consistently).

Decision letter (RSOS-191989.R0)

10-Mar-2020

Dear Dr Schaeffer

On behalf of the Editors, I am pleased to inform you that your Manuscript RSOS-191989 entitled "Metabolic rate in common shrews is unaffected by seasonal temperature, leading to reduction of

energetic costs through size reduction." has been accepted for publication in Royal Society Open Science subject to minor revision in accordance with the referee suggestions. Please find the referees' comments at the end of this email.

The reviewers and handling editors have recommended publication, but also suggest some minor revisions to your manuscript. Therefore, I invite you to respond to the comments and revise your manuscript.

- Ethics statement

- Data accessibility

If you wish to submit your supporting data or code to Dryad (<http://datadryad.org/>), or modify your current submission to dryad, please use the following link:
<http://datadryad.org/submit?journalID=RSOS&manu=RSOS-191989>

- Competing interests

- Authors' contributions

- Acknowledgements

- Funding statement

Because the schedule for publication is very tight, it is a condition of publication that you submit the revised version of your manuscript before 19-Mar-2020. Please note that the revision deadline will expire at 00.00am on this date. If you do not think you will be able to meet this date please let me know immediately.

Please note that Royal Society Open Science charge article processing charges for all new submissions that are accepted for publication. Charges will also apply to papers transferred to

Royal Society Open Science from other Royal Society Publishing journals, as well as papers submitted as part of our collaboration with the Royal Society of Chemistry (<https://royalsocietypublishing.org/rsos/chemistry>).

If your manuscript is newly submitted and subsequently accepted for publication, you will be asked to pay the article processing charge, unless you request a waiver and this is approved by Royal Society Publishing. You can find out more about the charges at <https://royalsocietypublishing.org/rsos/charges>. Should you have any queries, please contact openscience@royalsociety.org.

on behalf of Dr Cynthia Downs (Associate Editor) and Kevin Padian (Subject Editor)
openscience@royalsociety.org

Associate Editor Comments to Author (Dr Cynthia Downs):

Two expert reviews and I reviewed the manuscript “Metabolic rate in common shrews is unaffected by seasonal temperature, leading to reduction of energetic costs through size reduction.” A common ecological rule is that animals are larger in colder environments. This study investigated the energetic and behavior consequences of the paradoxical seasonal reductions in body size in the common shrew. The study shows that the combination of seasonal changes in behavior and morphology results in energy savings. Both reviewers felt that the study was scientifically sound and contributed to our understanding of thermal strategies and adaptations. The reviewers liked the integration of physiology and behavior and felt that it was represented a novel contribution to the literature on thermal strategies. I agree with the reviewers that parts of the introduction could be rearranged and edited to enhance the paper. I recommend making the hypothesis/aims of the study more salient and clarifying the description of the metabolic theory of ecology so that it more accurately reflects the literature on that topic. One reviewer’s comments focused on how conductance and body temperature differences might relate to the presented results; please integrate these concepts to enhance your discussion. Address the comments about the methods to increase the ability for replication of the study.

Reviewer comments to Author:
Reviewer: 1

Comments to the Author(s)

This paper investigates changes in metabolic rate in response to seasonal changes in temperature and phenotypic changes in body size and relative brain size to further elucidate the mechanisms behind Dehnel’s Phenomenon. The methods and data are sufficiently executed. The interpretation of the results are adequate.

Abstract: Generally good. I would suggest including a brief explanation for how the authors ruled out the following in the abstract “Body mass was the main driver of oxygen consumption, not the reduction of metabolically expensive brains”. For example, through the use of generalized mixed effects models. This is also unclear in the objectives paragraph 110-120.

Introduction: The authors state that MTE provides an important framework to tease apart the effect of body size and temperature as the main energetic costs for endotherms. This isn't exactly true. The temperature term in MTE (Brown et al. 2004) is included to accommodate ectotherms that vary in body temperature in response to ambient temperatures. The major extension of MTE and Scholander-Irving to accommodate impacts of temperature in endotherms to body size and metabolic rate comes from Fristoe et al. (2015 PNAS). That and Scholander showed how changes in environmental temperature influences changes in body temperature via metabolic rate and conductance. It seems the alternative hypothesis the authors are testing is that the mass-specific metabolic rate would be greater for body size in winter to offset heat lost to cold, although the authors do not find support for this. This may inform predictions in changes in body temperature, which the authors do not report here (and that's OK). But may wish to include in future studies. But I do think this section of the introduction should be revised to clearly explain the link between MTE and predictions.

Line 45: "It states that temperature" do the authors mean environmental or body temperature or both? See above explicit distinction in theory. This should be clear throughout when temperature is used, for example, in lines 58-59 is it body or environmental temp?

Line 85: Are the author's referring to metabolic scope?

Line 89: If *S. araneus* at 10 g has the same BMR as 25 g vole and two that of a similar sized white-toothed shrew, the does it have a higher body temperature, higher conductance or operate and colder environmental temperatures (e.g., Fristoe et al. 2015)? The authors do not report body temperature or conductance, but these links need to be clear in the context of the theory.

Line 113: again, you must be referring to environmental temperature unless you measured body temperature too?

Line 232: Please reference previously published studies you refer to.

Line 239: Please cite the previously published data source.

Paragraph 312-333: is very long. Consider breaking in to two.

Reviewer: 2

Comments to the Author(s)

General: I found the study as an interesting and thought-provoking. Authors continuously produce papers discussing morphological and physiological adaptations to seasonal changes of ambient temperatures in small, non-hibernating mammals (e.g. Dechmann et al. 2017, LaPoint et al. 2017, Lázaro et al. 2017 and 2018). Here, along with changes in body mass and skull size in common shrews across three seasons (summer, winter and spring), Authors estimated individual's energy consumption (absolute and mass-specific) together with five classes of behavior activity (rest, eat, drink, run and walk). And these two last measurements, metabolic rate and behavior, add new facts to the number of papers testing animals adaptations to challenging environmental factors. The demonstration of seasonal phenotypic plasticity in morphological, behavioral and metabolic traits is fundamental not only for better understanding one of winter strategy in small mammals called Dehnel's phenomenon, but also for the whole evolutionary physiology. From this perspective the study is important. However, I have several comments, which can help Authors make some corrections and improve the paper.

ABSTRACT

Page 2, line 30: This conclusion should be more specified. Activity is an important component of energy budget, and it is reflected in oxygen consumption. Please, re-phrase this sentence to make it less general.

INTRODUCTION

I have a little confusion with the expression of the aim of the study. If my meaning of Authors' intention is correct, scientific reasons for which the work was done are presented on page 5, lines 91-92 and lines 108-109. But now, they are hidden in the main text. Both arguments should be strongly emphasized together with a general description of methods used for such comparisons. It ones read then oxygen consumption was measured outside at ambient environmental temperatures (lines 111-112), but no word about a way in which a class of behavior was recording. Please, re-write lines 110-114 to make that paragraph (very essential for Introduction) more informative and scientific important.

Pages 5-6, lines 114-120: Re-think and make hypotheses more clear. In Discussion there is the statement 'we expected both body mass and environmental temperature to influence organismal metabolic rate' (page 12, lines 286-287). I can't see any detailed speculation about an effect of body mass reduction in shrews on their metabolic rate in that part of Introduction.

Page 4, line 84: Shouldn't be 'surface to volume ratio'?

METHODS

Page 6, line 140: Put information how long shrews have been waiting for the beginning of the experiments. The time course between capturing and experiments was the same for individuals from each season or not? If not, way?

Page 6, lines 141-144: I have mixed feelings with that information. It describes the method for incorporation digested food into metabolism and assessment of turnover rate of body fat, which was one of the aim in previous Authors' study (I suppose it is Keicher L., O'Mara M.T., Voigt C.C., Dechmann D.K. 2017. Stable carbon isotopes in breath reveal fast incorporation rates and seasonally variable but rapid fat turnover in the common shrew (*Sorex araneus*). *J. Exp. Biol.* 220, 2834-2841). Moreover, it looks like the same group of shrews was used in the previous paper and this work. And of course, a way like this is acceptable in results sharing and publishing, but I can see no scientific need for putting information like that here. Results on incorporation rates and fat turnover were published and they are not presented in actually reviewed paper. I suggest removing that information.

Page 6, line 144: Phrase 'two and seven days after capture' is confusing. It is not clear to which step of experiment it relates to. And again, does it describe the period of time before the start of oxygen consumption analyses? Please, specify it.

Page 7, line 147: Remove the phrase 'for 12 hours'. It is mentioned below, in line 164 (and refers to all shrews used in measurements of oxygen consumption).

Page 7, line 157: Please decide which form 'braincase' or 'brain case' is correct and apply it throughout the paper.

Page 7, lines 159-160: How long recovery time have been lasting before releasing shrews at their place of capture?

Page 8, lines 198-200: For each individual the 12-hour respirometry trial was conducted within a day, while behavior recording took 24 hours (including a night)? Am I right?

RESULTS

Page 10, line 240: Cite an appropriate paper at the end of the sentence.

Page 10, lines 247-248: Shouldn't be 'spring adults' instead of 'winter subadults'?

DISCUSSION

Page 12, lines 308-309: Cite an appropriate paper.

Page 13, line 319: Cite adequate figures.

Page 13, line 324: Put 'Figure 3E' instead of 'Figure 3D'.

OTHER COMMENTS:

Figure 1 and 2. The sample size for winter subadults and spring adults on both figures is the same (n=8 and n=6, respectively). Explain the difference in the number of summer juveniles presented on Figure 1 and 2 of (in Methods, consistently).

Author's Response to Decision Letter for (RSOS-191989.R0)

See Appendix A.

Decision letter (RSOS-191989.R1)

31-Mar-2020

Dear Dr Schaeffer,

It is a pleasure to accept your manuscript entitled "Metabolic rate in common shrews is unaffected by temperature, leading to lower energetic costs through seasonal size reduction." in its current form for publication in Royal Society Open Science. The comments of the reviewer(s) who reviewed your manuscript are included at the foot of this letter.

Please note that we require all authors to have active email addresses able to receive messages from the journal. Unfortunately, mmuturi@orn.mpg.de is not currently receiving messages. Please can you supply me with an alternative email address for Dr Muturi as soon as possible?

on behalf of Dr Cynthia Downs (Associate Editor) and Kevin Padian (Subject Editor)
openscience@royalsociety.org

Associate Editor Comments to Author (Dr Cynthia Downs):
Associate Editor
Comments to the Author:

This manuscript is well written, scientifically sound, and significantly advances advancing scientific knowledge about thermal strategies and thermal adaptations. I have reviewed it and it was not sent out for additional reviews. Thank you for clearly and satisfactorily addressing the previous reviews. I have one minor suggestion. Please add the statement "results not presented herein" to line 143.

Sincerely,
Cynthia Downs

Appendix A

Dear Dr. Downs and Dr. Dunn

We are, of course, excited about your conditional acceptance of our paper in Royal Society Open Science. We have gone through all the instructions in your email carefully, and added the required information, particularly also the end sections (new line numbers 344ff). We also thank you and the reviewers for the very welcome comments. Please find our responses below **in bold** as well as in the revised manuscript (as suggested we submit a version with and without tracked changes, the indicated line numbers refer to the tracked version). Please note that we have slightly changed the title as the previous one seemed a bit bumbling to us now.

We hope we have been able to clarify and address everything to your satisfaction and look forward to hearing from you.

Paul Schaeffer and co-authors

Associate Editor Comments to Author (Dr Cynthia Downs):

Two expert reviews and I reviewed the manuscript “Metabolic rate in common shrews is unaffected by seasonal temperature, leading to reduction of energetic costs through size reduction.” A common ecological rule is that animals are larger in colder environments. This study investigated the energetic and behavior consequences of the paradoxical seasonal reductions in body size in the common shrew. The study shows that the combination of seasonal changes in behavior and morphology results in energy savings. Both reviewers felt that the study was scientifically sound and contributed to our understanding of thermal strategies and adaptations. The reviewers liked the integration of physiology and behavior and felt that it was represented a novel contribution to the literature on thermal strategies.

>> Thank you for your positive review and the comments which we respond to in detail below

I agree with the reviewers that parts of the introduction could be rearranged and edited to enhance the paper. I recommend making the hypothesis/aims of the study more salient, and clarifying the description of the metabolic theory of ecology so that it more accurately reflects the literature on that topic.

>> We have tried to address these points as described in detail below. We have rewritten sections of the introduction to reflect reviewer comments, particularly as related to conductance and the specific hypotheses.

One reviewer’s comments focused on how conductance and body temperature differences might relate to the presented results; please integrate these concepts to enhance your discussion.

>> We have added a short comment on this point (new line numbers 292ff).

Address the comments about the methods to increase the ability for replication of the study.

>> We have addressed the comments in the methods to better explain how the animals were handled and give our rationale for why this should be included. We hope this addresses this concern (various line numbers, see responses to reviewers below).

Reviewer comments to Author:

Reviewer: 1

Comments to the Author(s)

This paper investigates changes in metabolic rate in response to seasonal changes in temperature and phenotypic changes in body size and relative brain size to further elucidate the mechanisms behind Dehnel's Phenomenon. The methods and data are sufficiently executed. The interpretation of the results is adequate.

Abstract: Generally good. I would suggest including a brief explanation for how the authors ruled out the following in the abstract “Body mass was the main driver of oxygen consumption, not the reduction of metabolically expensive brains”. For example, through the use of generalized mixed effects models. This is also unclear in the objectives paragraph 110-120.

>> There may be different schools with regards to this, but as the methods used are not what is the important message of the paper we have excluded the explanation of the statistical methods used to support the conclusion in both of these sections. These are clarified in the methods and results sections.

Introduction: The authors state that MTE provides an important framework to tease apart the effect of body size and temperature as the main energetic costs for endotherms. This isn’t exactly true. The temperature term in MTE (Brown et al. 2004) is included to accommodate ectotherms that vary in body temperature in response to ambient temperatures. The major extension of MTE and Scholander-Irving to accommodate impacts of temperature in endotherms to body size and metabolic rate comes from Fristoe et al. (2015 PNAS). That and Scholander showed how changes in environmental temperature influences changes in body temperature via metabolic rate and conductance. It seems the alternative hypothesis the authors are testing is that the mass-specific metabolic rate would be greater for body size in winter to offset heat lost to cold, although the authors do not find support for this. This may inform predictions in changes in body temperature, which the authors do not report here (and that’s OK). But may wish to include in future

studies. But I do think this section of the introduction should be revised to clearly explain the link between MTE and predictions.

>> You are correct that we have rather oversimplified this in the introduction, assuming too much familiarity with the seminal work of Scholander and Irving as well as the general principles of size, metabolism and conductance. We have added a section making it more clear how these are thought to be related and why reduction in body size may be an adaptation to winter in this animal living at the extremes of body size (new lines 43ff).

Line 45: "It states that temperature" do the authors mean environmental or body temperature or both? See above explicit distinction in theory. This should be clear throughout when temperature is used, for example, in lines 58-59 is it body or environmental temp?

>> you are correct, this is unclear. We mean environmental temperature and have changed this in the text.

Line 85: Are the author's referring to metabolic scope?

>> That is correct. We added that term to the text (new line 90).

Line 89: If *S. araneus* at 10 g has the same BMR as 25 g vole and two that of a similar sized white-toothed shrew, the does it have a higher body temperature, higher conductance or operate and colder environmental temperatures (e.g., Fristoe et al. 2015)? The authors do not report body temperature or conductance, but these links need to be clear in the context of the theory.

>> We have added these points to the text (new lines 93-94).

Line 113: again, you must be referring to environmental temperature unless you measured body temperature too?

>> again, you are correct and we have added "environmental"

Line 232: Please reference previously published studies you refer to.

>> This has been clarified to cite the previous work by Lazaro et al.

Line 239: Please cite the previously published data source.

>> This has been clarified to cite the previous work by Lazaro et al.

Paragraph 312-333: is very long. Consider breaking in to two.

>> done as suggested

Reviewer: 2

Comments to the Author(s)

General: I found the study as an interesting and thought-provoking. Authors continuously produce papers discussing morphological and physiological adaptations to seasonal changes of ambient temperatures in small, non-hibernating mammals (e.g. Dechmann et al. 2017, LaPoint et al. 2017, Lázaro et al. 2017 and 2018). Here, along with changes in body mass and skull size in common shrews across three seasons (summer, winter and spring), Authors estimated individual's energy consumption (absolute and mass-specific) together with five classes of behavior activity (rest, eat, drink, run and walk). And these two last measurements, metabolic rate and behavior, add new facts to the number of papers testing animals adaptations to challenging environmental factors. The demonstration of seasonal phenotypic plasticity in morphological, behavioral and metabolic traits is fundamental not only for better understanding one of winter strategy in small mammals called Dehnel's phenomenon, but also for the whole evolutionary physiology. From this perspective the study is important. However, I have several comments, which can help Authors make some corrections and improve the paper.

ABSTRACT

Page 2, line 30: This conclusion should be more specified. Activity is an important component of energy budget, and it is reflected in oxygen consumption. Please, re-phrase this sentence to make it less general.

>> We have modified this sentence to capture the idea that changes in the proportions of behaviours could impact overall energy demands.

INTRODUCTION

I have a little confusion with the expression of the aim of the study. If my meaning of Authors' intention is correct, scientific reasons for which the work was done are presented on page 5, lines 91-92 and lines 108-109. But now, they are hidden in the main text.

>> Yes, these set the stage for the questions asked in the final paragraph. We have clarified those questions to better follow from these points (new lines 113ff).

Both arguments should be strongly emphasized together with a general description of methods used for such comparisons. It ones read then oxygen consumption was measured outside at ambient environmental temperatures

(lines 111-112), but no word about a way in which a class of behavior was recording. Please, re-write lines 110-114 to make that paragraph (very essential for Introduction) more informative and scientifically important.

>> **We have rewritten this section to clarify the manner in which the experiments are done. As this is the Introduction, we don't go into great detail, but have made this point clearer (new line 119ff).**

Pages 5-6, lines 114-120: Re-think and make hypotheses more clear. In Discussion there is the statement 'we expected both body mass and environmental temperature to influence organismal metabolic rate' (page 12, lines 286-287). I can't see any detailed speculation about an effect of body mass reduction in shrews on their metabolic rate in that part of Introduction.

>> **We have rewritten the questions asked to try to capture the full extent of our predictions in this introductory section when describing the questions asked. We more explicitly ask whether size, temperature or behaviour is important in this system (new lines 113ff).**

Page 4, line 84: Shouldn't be 'surface to volume ratio'?

>> **Correct! Changed as suggested**

METHODS

Page 6, line 140: Put information how long shrews have been waiting for the beginning of the experiments. The time course between capturing and experiments was the same for individuals from each season or not? If not, way?

>> **No, as the same animals were also used in another experiment, where stable isotopes were collected from the breath, this time varied. However, as the animals were always kept at environmental conditions and the difference was a few days at most, we assume it did not influence energy consumption (new line 147ff).**

Page 6, lines 141-144: I have mixed feelings with that information. It describes the method for incorporation digested food into metabolism and assessment of turnover rate of body fat, which was one of the aim in previous Authors' study (I suppose it is Keicher L., O'Mara M.T., Voigt C.C., Dechmann D.K. 2017. Stable carbon isotopes in breath reveal fast incorporation rates and seasonally variable but rapid fat turnover in the common shrew (*Sorex araneus*). J. Exp. Biol. 220, 2834-2841). Moreover, it looks like the same group of shrews was used in the previous paper and this work. And of course, a way like this is acceptable in results sharing and publishing, but I can see no scientific need for putting information like that here. Results on incorporation rates and fat turnover were published and they are not presented in actually reviewed paper. I suggest removing that information.

>> **While we agree that there is no presentation of these data, nor any link between the data presented and this 'breath sampling' protocol, we include it solely to be clear about how the animals were handled prior to the MR measurements presented in this work. We do not consider these to be particularly relevant for the outcomes, nor do we feel that any small handling differences were likely to have contributed to the outcomes, but we prefer to leave this for the sake of full transparency.**

Page 6, line 144: Phrase 'two and seven days after capture' is confusing. It is not clear to which step of experiment it relates to. And again, does it describe the period of time before the start of oxygen consumption analyses? Please, specify it.

>> **We have clarified this section that describes how the animals were used in the experiments described in [33], with a small difference in handling between seasons (see also above).**

Page 7, line 147: Remove the phrase 'for 12 hours'. It is mentioned below, in line 164 (and refers to all shrews used in measurements of oxygen consumption).

>> **done as suggested**

Page 7, line 157: Please decide which form 'braincase' or 'brain case' is correct and apply it throughout the paper.

>> **changed to braincase throughout**

Page 7, lines 159-160: How long recovery time have been lasting before releasing shrews at their place of capture?

>> **We now note that all were released within 2 hours after completion of the experiments.**

Page 8, lines 198-200: For each individual the 12-hour respirometry trial was conducted within a day, while behavior recording took 24 hours (including a night)? Am I right?

>> **No each animal was recorded during 12 hours for behaviour, too, either at night or during the day. We slightly rephrased this to make it clearer (new line 200).**

RESULTS

Page 10, line 240: Cite an appropriate paper at the end of the sentence.

>> **added two citations**

Page 10, lines 247-248: Shouldn't be 'spring adults' instead of 'winter subadults'?

>> **well spotted, of course you are right. Changed accordingly.**

DISCUSSION

Page 12, lines 308-309: Cite an appropriate paper.

>> **We have added the citation (47).**

Page 13, line 319: Cite adequate figures.

>>**inserted several Figure citations**

Page 13, line 324: Put 'Figure 3E' instead of 'Figure 3D'.

>> **We have corrected this.**

OTHER COMMENTS:

Figure 1 and 2. The sample size for winter subadults and spring adults on both figures is the same (n=8 and n=6, respectively). Explain the difference in the number of summer juveniles presented on Figure 1 and 2 of (in Methods, consistently).

>> **An individual that was captured but not used in this study was included in the plot. This has been corrected and all other sample size checked for completeness. New Figures 1 and 3 are included in the revision**